# A Study of the Sulfidation Behavior on Palladium-Coated Copper Wire with a Flash-Gold Layer (PCA) after Wire Bonding

**Kuan-Jen Chen** [1] , **Fei-Yi Hung** [2,*] and **Chia-Yun Chang** [2]

1   Instrument Center, National Cheng Kung University, Tainan 701, Taiwan
2   Department of Materials Science and Engineering, National Cheng Kung University, Tainan 701, Taiwan
*   Correspondence: fyhung@mail.ncku.edu.tw; Tel.: +886-6-2757575-62950

**Abstract:** Palladium-coated copper wire with a flash-gold layer (PCA) is an oxidation-resistant fine wire that simultaneously has the properties of palladium-coated copper wire (PCC) and gold-coated copper wire. This research used an extreme sulfidation test to compare corrosion resistance between the PCC and PCA wires. In addition to closely examining the morphology of the wires, the internal matrix after the sulfidation test is also discussed. In doing so, the PCA wire was bonded onto the aluminum pads and the sulfidation test was conducted. Then, we observed its morphology and elemental distribution and found that the flash-gold layer of the PCA wire effectively enhanced resistance to sulfidation corrosion. Because the copper ball had an alloying effect on the ball bonding, it produced different shapes of sulfide after the sulfidation test. The degree of corrosion on the wedge bond was different because of the presence or absence of the coated layer. In contrast, the flash-gold layer of the PCA wire enhanced the bonding force and retained low resistance characteristics after the sulfidation test.

**Keywords:** copper wire; flash-gold layer; wire bonding; sulfidation

## 1. Introduction

Due to the high price of gold (Au), lower-cost copper (Cu) and aluminum (Al) wires have gradually become the main bonding wires [1–4]. Ultrasonic wedge bonding of Al wire is the preferred method for electrical connections between devices and chip carriers [5]. Unfortunately, Al wire is easily oxidized and is not conducive to forming free air balls (FABs), which deteriorates the reliability of the wire bonding. Shortcomings of Cu wire include its high hardness and easy oxidation, however, it can be improved by annealing and coating with an antioxidant layer [6]. Commonly used Cu-based wires include the palladium (Pd)-coated Cu wire (PCC) and the Au-coated Cu wire. Although both can improve oxidation, some reliability problems remain with the processes. For example, if the Pd-coated layer is too thick, the wire will break during the drawing process [7]. A problem occurs after the electronic flame-off (EFO) process, where the Pd atoms segregate near the neck zone, and thus reduce the neck strength [8,9]. In addition, if the Au-coated layer is too thick, the free air ball (FAB) will be spear-like (lemony shape) and an Au-Al intermetallic compound will form after bonding [10]. Therefore, this study combines the advantages of PCC wire and Au-coated Cu wire. First, a Pd layer was coated on the surface of the Cu wire, after which a flash-Au layer was coated to form the Pd-coated Cu wire with flash-Au (PCA wire). Such coatings can improve the corrosion resistance, provide lubricity during the drawing process, enhance the bonding strength, reduce solidification segregation during FAB forming, and reduce the formation of intermetallic compound (reduce electrical resistance).

The study compares the sulfidation corrosion resistance between the PCC wire and the PCA wire [6], and includes a discussion of the morphology, elemental distribution, and bonding strength of

the first and second bonds of PCA wire after the sulfidation test [11]. At present, in the literature, reports of double-coated wire are very scarce. In response, a Cu wire is coated with a Pd layer by electroplating and then a flash-Au layer on the PCC wire to form PCA. This process is an innovative technique, and it uses sulfidation experiments to accelerate the evaluation of wire corrosion characteristics. Moreover, it is the world's first method for fine wire research, and the data provides a reference which is relevant to the Cu process for selecting the molding compound type and the long-term reliability experiment.

## 2. Materials and Methods

In this study, the diameter of the PCA wire is 18 µm, where the thicknesses of the Pd-coated layer and the flash-Au layer are approximately 70 nm (2 wt%) and 10 nm, respectively. All the chemicals were purchased from NICHE-TECH GROUP LIMITED (NTG, Hong Kong). For the sulfidation corrosion test, the PCC and PCA wires were treated in a stove at 140 °C for 20 min to 4 h. Subsequently, the surface morphology and the cross-sectional matrix of the wire specimens were examined using a dual beam focused ion beam with a scanning electron microscopy (FIB-SEM, Quanta 3D FEG, FEI, OR, USA) [12]. After the sulfidation test, the depth distributions of each element for the PCC and PCA wires were analyzed using an auger electron spectroscopy (AES, Microlab 350, Thermo Fisher Scientific, Runcorn, UK) [13].

The FABs of the PCC and PCA wires were formed using the EFO process, after which the FABs were bonded to the Al pad to perform the first bond (ball bond), and then moved to the other site to complete the second bond (wedge bond). The microtensile tester (TD-121, JobHo Technology, Taichung, Taiwan) was conducted to the wire and the ball bond after the sulfidation test, with the tensile velocity set at $5 \times 10^{-3}$ s$^{-1}$ [14], and the test length was 50 mm and 10 mm, respectively. Each datum is the average of at least five specimens. To clarify the effects of the sulfidation on the electrical properties of the PCC and PCA wires, a direct current was applied to the sulfured wires and ball bonds starting at 0 A and increased in steps of 0.05 A until the fusion. The fracture surface of the sulfured ball bonds was also obtained using FIB-SEM after the tensile test.

## 3. Results

### 3.1. Surface Morphologies and Cross Sections of the Wires

Figure 1a,b shows the surface morphology and cross-section images of the PCC and PCA wires after the sulfidation of 4 h. We observed that most of the PCC wire was eroded and there was a large amount of sulfide growth on the surface. In contrast, the surface of the PCA wire was only slightly corroded, with only a small amount of sulfide growth. Our observations of the cross-section images, showed that the sulfide layer near the surface of the PCC wire was thicker than that of the PCA wire. Notably, the PCC wire had large and more numerous voids in the subsurface. The presence of the voids is attributed to the corrosive sulfur that reacts with Cu to form a sulfide, and thus consumes the Cu atoms of the wire [15,16]. To understand the diffusion behavior of each atom, the depth profile of element distribution was obtained by AES, as shown in Figure 1c,d. In the case where the thickness of the Pd-coated layer (~70 nm) was similar, the sulfur elements in the PCC wire penetrated from the surface to a depth of about 30 nm. By contrast, the sulfur elements in the PCA wire, with the flash-Au layer on the surface, only penetrated to a depth of 20 nm (Figure 1d). This result indicates that the flash-Au layer can protect the PCC wire from sulfur by limiting the growth of copper sulfide. Briefly, the PCA wire has better resistance to sulfur corrosion.

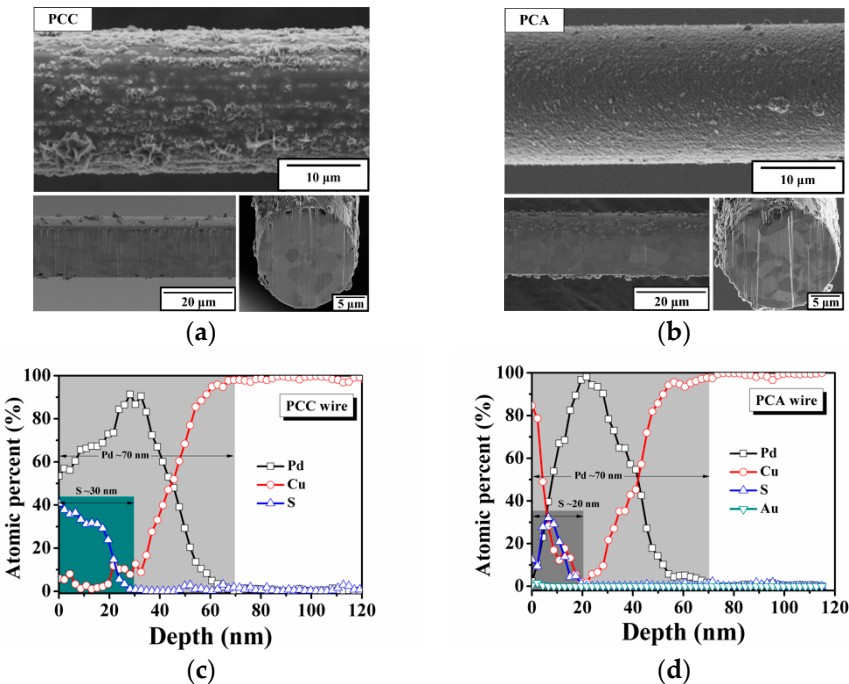

**Figure 1.** (**a**,**b**) Surface morphologies, cross-section images, and (**c**,**d**) depth profile of the PCC and PCA wires after the sulfidation for 4 h.

### 3.2. Mechanical and Electrical Properties of the Wires

In order to understand the effect of sulfidation on the mechanical properties of the wires, the tensile test results for the PCC and PCA wires after the sulfidation are shown in Figure 2. The tensile properties of the PCA wire were essentially lower than those of the PCC wire, which is associated with one more flash-Au process than the PCA wire. The fracture load and ductility of both wires slightly decreased as the sulfidation duration increased, but the PCA wire decreased less. Notably, the ductility of the PCC wire decreased significantly after sulfidation for 20 min, whereas, that of the PCA wire showed significant deterioration after sulfidation of 4 h (Figure 2b). The contribution of the flash-Au layer on the PCA wire can effectively retard the deterioration of the wire for at least 2 h.

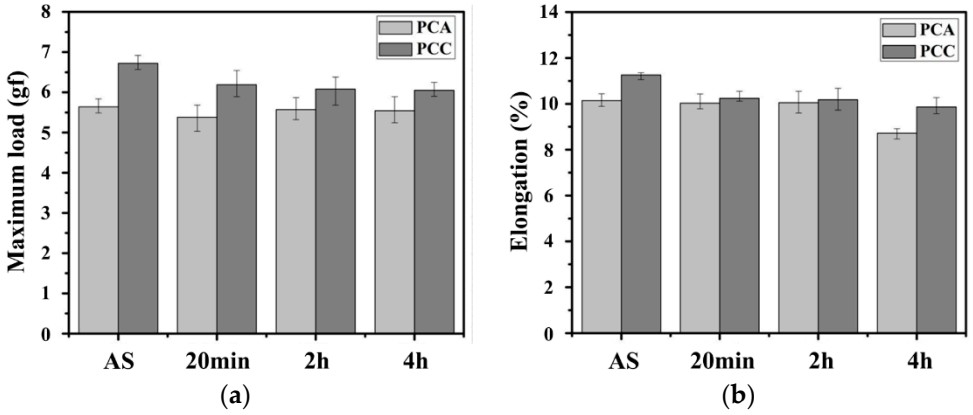

**Figure 2.** Tensile properties of PCA and PCC wires after sulfidation: (**a**) Maximum load, and (**b**) elongation.

The electrical properties of the PCC and PCA wires before and after the sulfidation test were examined (Figure 3). Before the sulfidation test, all wires (pure Cu, PCC, and PCA) were affected by electromigration and bias-induced Joule heat as the current increased, causing the wire resistance to

rise (Figure 3a). The wires would melt as the current was increased, and thus acquired a critical fusion current (CFC). The current–voltage (I-V) curves of the PCC and PCA wires almost overlap, and that of the pure Cu wire exhibits different trends at low current and high current. The surface of the pure Cu wire was easily oxidized at the low current (0.05–0.15 A), affecting the electrical resistance of the wire. At the high current (0.25–0.35 A), the grain size in the pure Cu wire increased by the Joule heat [17], causing the electrical resistance to slightly decrease. The PCC and PCA wires have similar electrical properties, indicating that the Pd-coated layer and the flash-Au layer did not participate in the electron transport behavior [6]. In the case of the short length of the 50 mm sulfured wires (Figure 3b), the starting voltage of the PCC wire was higher than that of the PCA wire due to the PCC wire having more sulfides and voids near the surface after the sulfidation, increasing the electrical resistance of the PCC wire. As the sulfidation duration and the length of the tested wire was extended to 4 h and 300 mm, respectively, (Figure 3c), the starting voltage was not significantly different. Notably, the electrical resistance of the wires increased as the current increased, and the difference in resistance of the PCC wire before and after the sulfidation was greater than that of the PCAwire. This result is attributed to the flash-Au layer which served as the protection layer to retard the sulfur corrosion [14]. The effects of the sulfidation on the surface morphology, mechanical, and electrical properties of the PCA wire were small. This indicates that the flash-Au layer of the PCA wire can effectively protect the wire and enhance the resistance to the sulfidation.

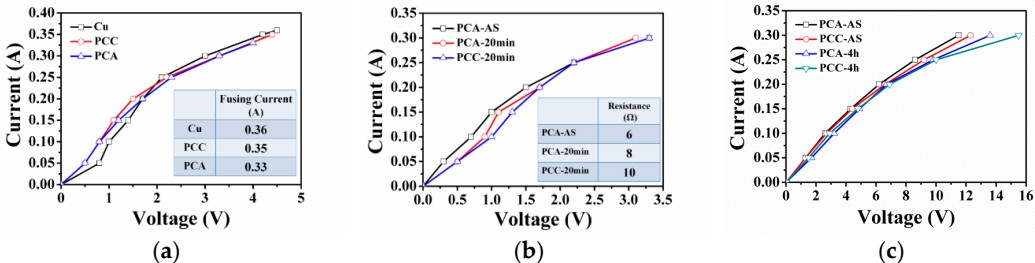

**Figure 3.** Electrical properties of PCC and PCA wires (**a**) before and (**b**,**c**) after sulfidation.

### 3.3. Bonding Strength and Electrical Properties

The morphologies of the ball and wedge bonds of the PCA wire after sulfidation for 20 min and 4 h are shown in Figure 4. It shows that the ball bond has three types of sulfide after sulfidation (Figure 4a), namely, (1) leaf-like sulfide at the bonding site near the Al substrate, (2) particle sulfide at the peripheral area, and (3) hairy sulfide at the central region. By the proportion of elements, the leaf-like sulfide is $Cu_2S$ (hexagonal phase) [18], while the particle and hairy sulfides are CuS (HCP) [19]. The slow diffusion of Cu atoms reacted with the sulfur, causing the formation of the leaf-like $Cu_2S$ microstructure which spread out from the interface between the ball bond and the Al substrate [20,21]. Peripheral zones with a large contact area of sulfur vapor caused a rapid reaction between Cu and S and formation of spherical CuS crystals [22]. The higher ratios of Pd and Au in the central region retarded the sulfidation corrosion, and thus formed the hairy CuS crystals. For the wedge bond, the solidification segregation problem of the FAB did not occurr, only granular and hairy sulfides were formed. Due to a shorter sulfidation duration, most of the sulfides were hairy crystals, and the wire surface area was still smooth. After increasing the sulfidation duration (Figure 4b), the spherical CuS crystals at the peripheral zone of the ball bond were denser, while the hairy CuS crystals changed into granular crystals. A large amount of the sulfides formed on the ball bond, which may affect its mechanical and electrical properties. For the 4 h sulfured wedge bond, the sulfides completely covered the wire surface except the end of the fish tail. Note that a high concentration of Pd and a lower concentration of S were detected in the "D" position. This result is attributed to the fact that the Pd-coated layer was squeezed to the end of the fish tail during the wedge bonding process, and thus the Pd-rich area provided good resistance to sulfidation corrosion. Accordingly, the wedge bond of the coated wire should reduce the bond force and drag conditions.

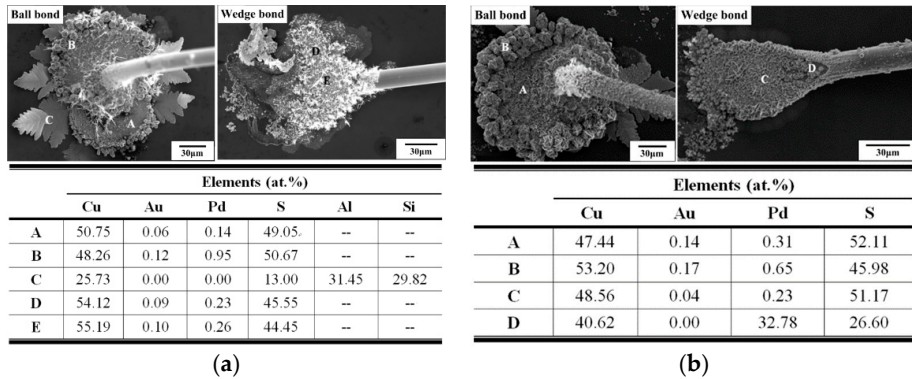

| Elements (at.%) | | | | | |
|---|---|---|---|---|---|
| | **Cu** | **Au** | **Pd** | **S** | **Al** | **Si** |
| **A** | 50.75 | 0.06 | 0.14 | 49.05 | -- | -- |
| **B** | 48.26 | 0.12 | 0.95 | 50.67 | -- | -- |
| **C** | 25.73 | 0.00 | 0.00 | 13.00 | 31.45 | 29.82 |
| **D** | 54.12 | 0.09 | 0.23 | 45.55 | -- | -- |
| **E** | 55.19 | 0.10 | 0.26 | 44.45 | -- | -- |

| Elements (at.%) | | | |
|---|---|---|---|
| | **Cu** | **Au** | **Pd** | **S** |
| **A** | 47.44 | 0.14 | 0.31 | 52.11 |
| **B** | 53.20 | 0.17 | 0.65 | 45.98 |
| **C** | 48.56 | 0.04 | 0.23 | 51.17 |
| **D** | 40.62 | 0.00 | 32.78 | 26.60 |

(**a**)                                     (**b**)

**Figure 4.** Morphologies of PCA wire bonding after different sulfidation durations: (**a**) 20 min and (**b**) 4 h.

To evaluate the corrosion resistance after the wire bonding, the ball bond of the PCA wire after the sulfidation was subjected to the tensile test, as shown in Figure 5a. By increasing the duration of the sulfidation, the fracture load of the sulfured ball bonds is similar to that of unsulfured ball bond. The sulfidation mainly affected the surface of the wire, and then formed a protective layer of hard sulfide on the wire surface [14]. Observations of the fracture surface (Figure 5b) show that the fracture of both the ball bonds occurred at the heat affected zone near the neck. Observations of the fracture location show that the thickness of the sulfide increased with an increase in the sulfidation duration. Although the sulfidation reaction consumes the Cu atoms of the wire which deteriorates the wire strength, the formation of sulfide on the ball bond surface increases its strength. The brittleness effect of the sulfide was not significant, and therefore the tensile strength of the sulfured ball bond did not change significantly.

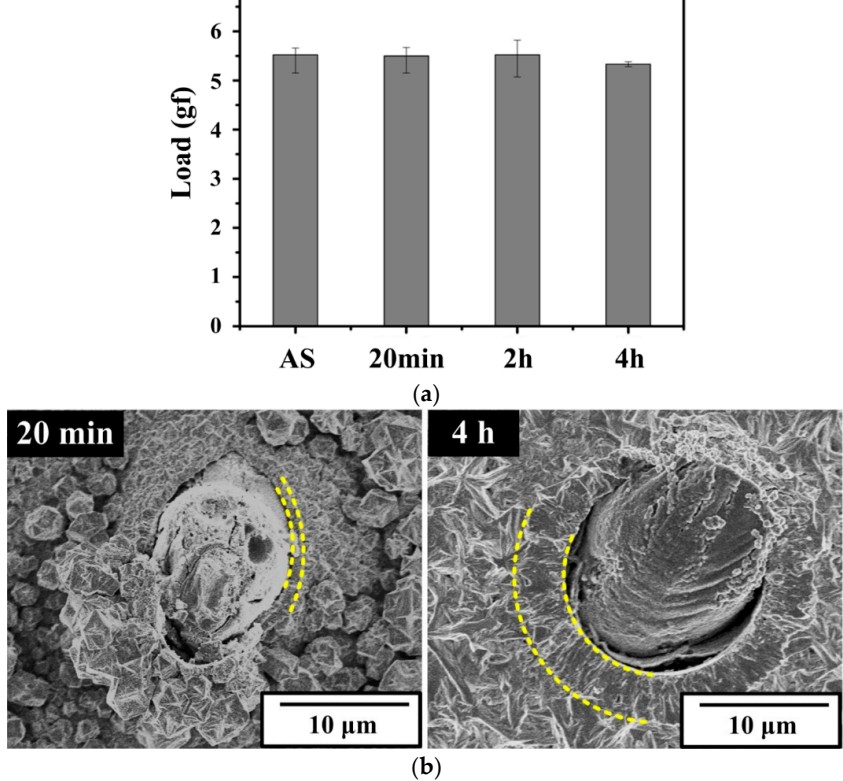

**Figure 5.** (**a**) Fracture load and (**b**) fracture surface of PCA ball bond wire after sulfidation.

The I-V measurements of the ball bond of the PCA wires after the various sulfidation durations are shown in Figure 6. As the sulfidation duration increased, the I-V curves shifted toward high voltage, indicating that the electrical resistance of the ball bond increased. The proportion of Cu on the ball bond surface increased after the FAB forming, and thus increased the reaction between the Cu and the sulfur vapor. Therefore, the thickness of the sulfide increased, reducing the current path in the ball bond wire which deteriorated the electrical resistance. The flash-Au layer of the PCA wire can enhance resistance to the sulfidation corrosion and can maintain a longer operating life. In brief, the PCA wire can improve the reliability for application in molding compounds.

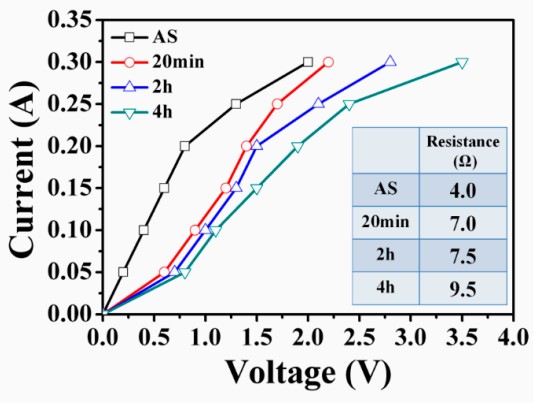

**Figure 6.** Electrical properties of the PCA ball bond wire after sulfidation.

*3.4. Limitations and Suggestion of this Study*

Limitations: Because the wire specimen was small, the sulfide could not be detected by X-ray diffraction (XRD). As such, it was difficult to explore whether the bonding interface was also affected by the penetration of sulfur. The circuit resistance of the wire specimens could not be measured after the sulfidation test. Suggestion: It is important to investigate the residual coating of the wedge bonding joints and establish a mechanism for sulfidation corrosion in the future. In addition, the Al pad was commonly used to connect with metal wires to reduce the costs of optoelectronic devices. In advanced devices, the electrode material can be changed to the Au pad, improving the electrical properties and reliability.

**4. Conclusions**

The flash-Au layer of PCA wire provides good resistance to sulfidation corrosion. After the ball bonding and sulfidation, there were three types of sulfides, namely, leaf-like $Cu_2S$, hairy CuS, and granular CuS. Further, the hairy sulfide gradually grew and became granular sulfides with an increase in the sulfidation duration. The wedge bonding joint only had hairy and granular sulfides. Regardless of the wire or ball bond, the strength had a tendency to decrease after the sulfidation test. Nevertheless, the decreasing strength tendency of the PCA wire was relatively small and the flash-Au layer provided protection for low electrical resistance. This study confirmed that the PCA wire could maintain good reliability in different molding compound systems.

**Author Contributions:** Investigation, C.-Y.C.; writing—original draft preparation, F.-Y.H.; writing—review and editing, K.-J.C.; project administration, F.-Y.H.

**Funding:** This research was funded by the MINISTRY OF SCIENCE AND TECHNOLOGY (MOST), grant number 107-2221-E-006 -012 -MY2.

**Acknowledgments:** The authors are grateful to the Instrument Center of National Cheng Kung University.

**Conflicts of Interest:** The authors declare no conflict of interest. The funders had no role in the design of the study; in the collection, analyses, or interpretation of data; in the writing of the manuscript, or in the decision to publish the results.

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
