# Peer review of "A Study of the Sulfidation Behavior on Palladium-Coated Copper Wire with a Flash-Gold Layer (PCA) after Wire Bonding"

_electronics, doi:10.3390/electronics8070792_

Round 1

Reviewer 1 Report

Comparison of  the sulfidation corrosion resistance between the PCC and the PCA wire was made. Authors discussed the morphology, elemental distribution and bonding strength of the first and  second bonds of PCA wire after the sulfidation test. Selection of the molding compound type for long-term reliability is explained in their experiments. Variations in the strength  of the PCA wire were noted. The flash-Au layer is found to  provide protection for low electrical resistance. Authors experimentally confirmed that the PCA wire could maintain good reliability in different molding compound systems.

Author Response

Response to Reviewer’s Comments

Reviewer 1

1.     Comparison of the sulfidation corrosion resistance between the PCC and the PCA wire was made. Authors discussed the morphology, elemental distribution and bonding strength of the first and second bonds of PCA wire after the sulfidation test. Selection of the molding compound type for long-term reliability is explained in their experiments. Variations in the strength of the PCA wire were noted. The flash-Au layer is found to provide protection for low electrical resistance. Authors experimentally confirmed that the PCA wire could maintain good reliability in different molding compound systems.

Our response:

The English language and style have been revised in throughout the paper.

Reviewer 2 Report

This paper presents palladium-coated copper wire with flash-gold layer (PCA) to investigate sulfidation behavior after bonding. Interestingly, the PCA wire can effectively enhance the bonding force and resistance in the sulfidation corrosion. Considering the importance of wire bonding in electronics applications, this paper would be of interest to researchers in electrical, mechanical, and material engineering. However, some comments below must be addressed for publication in ‘Electronics’.

1. Detailed experimental methods including fulling test, elongation test, and electrical test should be discussed in the manuscript.

2. Aluminum (Al) is one of popular bonding wires. In Introduction section, authors should discuss Al wire for bonding applications in electronics including following references. [J. Phys. D: Appl. Phys 49 (2016) 285109; J. Electron. Mater. 35 (2006) 433–42; Appl. Surf. Sci. 387 (2016) 280]

3. The material of substrate (pad) can also affect to the mechanical, electrical, and chemical properties of the bonded wire. Authors need to discuss how the results would be changed if the substrate is changed from aluminum to other metal materials.

Author Response

Response to Reviewer’s Comments

Reviewer 2

1.     Detailed experimental methods including fulling test, elongation test, and electrical test should be discussed in the manuscript.

Our response:

In tensile test and electrical measurement, the experimental methods have been added in the revision paper.

2.     Aluminum (Al) is one of popular bonding wires. In Introduction section, authors should discuss Al wire for bonding applications in electronics including following references. [J. Phys. D: Appl. Phys 49 (2016) 285109; J. Electron. Mater. 35 (2006) 433–42; Appl. Surf. Sci. 387 (2016) 280]

Our response:

Ultrasonic wedge-bonding of Al wire is the preferred method for electrical connection between devices and chip carriers [5]. Unfortunately, Al wire is easily oxidized, and is not conducive to form a free air ball (FABs), deteriorating the reliability of wire bonding. These sentences have been added in the introduction section. Also, the literature related to Al bonding wire has been cited in this revision manuscript.

3.     The material of substrate (pad) can also affect to the mechanical, electrical, and chemical properties of the bonded wire. Authors need to discuss how the results would be changed if the substrate is changed from aluminum to other metal materials.

Our response:

Al pad was commonly used to connect with metal wires to reduce the costs of optoelectronic devices. In advanced devices, the electrode material can be changed to Au pad, improving the electrical properties and reliability. These sentences have been added in the revision paper.

Reviewer 3 Report

1- Please proof read the paper for language typos.

2- line 52: Justify your choice of 140°C, 20 min and 4 h

3- line 71: provide time analysis to justfiy why you stopped at 4h.

4- line 78: provide scientific arguments to support your claim of the xistence of void.

5- line 80 to 82: this sentence is not clear. In addition (1) precise the thickness. (2) refers to figure, (3) add arrows in figure to support your claims.

6- Figure 2: how many times did you repeat the experiment? 

7- Explain your error bars.

8- line 106: why grain size increaed? explain the theory behind this to support your claim. You mentioned joule heat but is it supported by experiments or other paper. I did not see a curve of temperature versus grain size.

9- line 113: the tarting voltage ....: this confirms that there are concerns about the choice of 4h. Why 4?

10- line 115: this results is ... : please justify.

11- line 117: provide a proof. Are you referring to Fig 2?

12- it is not clear why at fig 4a we Al and Si and not at fig 4b.

13- line 142: the sulfide completely....: highlight this in the table and fig.

14- line 14:accordongly...: what is the base of this claim?

15- fig 5: how did you calculate your error bars and please mention ho many times did you repeat the experiment.

16- Fig 6: resistances is increasing and it is not negligeable!!! comment on this. What is the benefit of this then?

Author Response

Response to Reviewer’s Comments

Reviewer 3

1.     Please proof read the paper for language typos.

Our response:

Throughout the paper, the English language has been revised.

2.     line 52: Justify your choice of 140°C, 20 min and 4 h

Our response:

To evaluate the corrosion resistance of the wire and the reliability of the bonding, the sulfur was heated above the melting point (~115°C) to form sulfur vapor in the sulfidation corrosion test.

3.     line 71: provide time analysis to justify why you stopped at 4h.

Our response:

According to our previous report [11], the surface of the PCA wire did not show any significant change after the sulfured for 2 h. The PCA wire showed only the initial characteristic of corrosion until the sulfidation time increased to 4 h. In addition, the ductility of the PCA wire showed significant deterioration after the sulfidation of 4h. Therefore, the sulfidation time in present study was set to stop at 4h.

4.     line 78: provide scientific arguments to support your claim of the existence of void.

Our response:

According to previous reports [15-16], the sulfur easily reacted with copper to form copper sulfide. In present study, the sulfur vapor passed through the coating layer (Au and Pd) to react with Cu (consumed the Cu atoms of the wire) to form copper sulfide, thus formed the voids at the interface between the sulfide and the Cu wire. The related references have been cited in the revision paper.

[15]  VAN OOIJ, W.J. The role of XPS in the study and understanding of rubber-to-metal bonding. Surf. Sci. 1977, 68, 1-9.

[16]  Ozawa, K.; Kakubo, T.; Shimizu, K.; Amino, N.; Mase, K.; Komatsu, T. High-resolution photoelectron spectroscopy analysis of sulfidation of brass at the rubber/brass interface. Appl. Surf. Sci. 2013, 264, 297-304.

5.     line 80 to 82: this sentence is not clear. In addition (1) precise the thickness. (2) refers to figure, (3) add arrows in figure to support your claims.

Our response:

In the case where the thickness of the Pd-coated layer (~70 nm) was similar, the sulfur element in the PCC wire could penetrate from the surface to a depth of about 30 nm. By contrast, for the PCA wire with the flash gold layer on the surface, sulfur elements only penetrated into the depth of 20nm (Fig. 1(d)). In addition, the arrows have been added in the Fig. 1(c) and (d).

6.     Figure 2: how many times did you repeat the experiment?

Our response:

In tensile test, each datum is the average of at least five specimens.

7.     Explain your error bars.

Our response:

In tensile test, the error bar of the PCA wire has a lower standard error compared to the PCAA wire, indicating that the flash-Au layer can improve the reliability of bonding wire.

8.     line 106: why grain size increased? explain the theory behind this to support your claim. You mentioned joule heat but is it supported by experiments or other paper. I did not see a curve of temperature versus grain size.

Our response:

The grain size of the given wire increased with increasing bias duration, which attributed to the bias-induced Joule heat promoted the grain to grow. The similar result has been reported in our previous report [17]. The small grains were disappeared after the current test and many larger grains with equal size to the wire diameter were formed, which attributed to the current-induced dynamic grain growth. This previous report has been cited in the revision manuscript.

[17]  Hsueh, H.W.; Hung, F.Y.; Lui, T.S.; Chen, L.H. Effect of the direct current on microstructure, tensile property and bonding strength of pure silver wires. Microelectron. Reliab. 2013, 53, 1159-1163.

9.     line 113: the starting voltage ....: this confirms that there are concerns about the choice of 4h. Why 4?

Our response:

According to our previous report [14] and the tensile test results (Fig. 2), the surface morphology and tensile properties of the wire showed a significant change until the sulfidation time increased to 4h. Therefore, the effects of the sulfidation times on the electrical properties of the wires were controlled from 20min to 4h.

10.  line 115: this results is ... : please justify.

Our response:

The results of the previous report [14], the cross-sections images and SIMS analysis (Fig. 1) confirmed that the flash-Au layer was served as the protection layer to retard the sulfur corrosion. The Ref. [14] has been cited in the revision paper.

11.  line 117: provide a proof. Are you referring to Fig 2?

Our response:

According to Fig. 1 and Fig. 2, it confirmed that the flash-Au layer was served as the protection layer to retard the sulfur corrosion. Therefore, the PCA wire was smaller influences in the sulfidation environments.

12.  it is not clear why at fig 4a we Al and Si and not at fig 4b.

Our response:

In the initial stage of the sulfidation test (20 min), the ball bond has three types of copper sulfide. At “C” position, the leaf-like sulfide was formed at the bonding site near the Al substrate. Therefore, the Al and Si elements can be detected (Fig. 4(a)). After sulfided for 4 h, a large amount of the sulfides formed on the ball bond and covered with Al substrate. In addition, the caption of Fig. 4 has been revised.

13.  line 142: the sulfide completely....: highlight this in the table and fig.

Our response:

Note that a high concentration of Pd and a lower concentration of S were detected in the “D” position. This sentence has been added in the revision paper.

14.  line 145: accordingly...: what is the base of this claim?

Our response:

The coated layer (Pd) of the wire was squeezed to the end of the fish tail during the wedge bonding process, making more Pd atoms to concentrate in the neck. Therefore, the coated wire in the wedge bonding process can reduce the bond force and drag conditions to reduce the coated layer was squeezed to the neck.

15.  fig 5: how did you calculate your error bars and please mention how many times did you repeat the experiment.

Our response:

In tensile test, each datum is the average of at least five specimens, and then used average value, maximum value and minimum value to acquire error bars.

16.  Fig 6: resistances are increasing and it is not negligible!!! Comment on this. What is the benefit of this then?

Our response:

The copper sulfide was more easily formed on the ball bond surface due to the ball bond surface has a more proportion of Cu after the FAB forming. With increasing the sulfidation duration, the sulfur continually reacted with Cu to form copper sulfide, and thus consumed the Cu atoms in the ball bond wire, further causing the formation of the voids. The formation of the voids reduced the current path in the ball bond wire, and thus deteriorated its electrical resistance.

Round 2

Reviewer 2 Report

The authors addressed the reviewer’s comments/concerns well.  The overall contents and conclusions are solid and well supported by additional figures and comments.  I would like to recommend this manuscript to be accepted by Electronics without further revision.